# Field Geological Investigations and Stability Analysis of Duanjiagou Landslide

**Xingang Wang** [1,2]ⓘ**, Tangdai Xia** [1,]*****, Longju Zhang** [2]**, Min Gao** [1] **and Kang Cheng** [1]

[1]    College of Architectural Engineering and Zhejiang University, Hangzhou 310012, China;
       11712036@zju.edu.cn (X.W.); 201812013@zju.edu.cn (M.G.); chengkang@zju.edu.cn (K.C.)
[2]    College of Water Conservancy and Architectural Engineering, University of Tarim, Alar 843300, China;
       11612065@zju.edu.cn
*    Correspondence: xtd@zju.edu.cn

**Abstract:** This paper analyses the stability of the Duanjiagou landslide on the Bazhong to Guangan Expressway K134–K135 segment in China. The Duanjiagou landslide took place on 4 November 2015. In order to discover the cause of the landslide, we carried out field geological investigations. The indoor physical property experiments were performed by taking the undisturbed soil sample from the borehole cores. To study the strength of the soil, we carried out a saturation direct shear test and saturation residual shear test on sliding zone soil samples. According to the physical properties of soil and the saturated shear strength parameters of sliding zone soil, the stability was analyzed by the landslide force transmission method and numerical simulation method. The results showed that in the initial sliding stage, the safety factor obtained by using the average value of saturated shear strength parameters was in good agreement with the field observation situation. The landslide was at an unstable state. The softening of soil and roadbed excavation at the foot of the slope are the main reasons for landslides.

**Keywords:** Duanjiagou landslide; saturated direct shear test; stability analysis; safety factor

## 1. Introduction

Landslides are one of the most common and effective ways of shaping the surface morphology [1]. It exists on mountainous and hilly areas all over the world. A landslide is a kind of natural disaster caused by a series of processes, often affecting human activities and building environments, and causing catastrophic consequences. The previous research shows that landslides cause more casualties in developing countries such as India, China, Nepal, Peru, Venezuela and the Philippines [2–4]. A landslide is related to seasonal heavy rainfall, erosion excavation at the foot of the slope, unreasonable surcharge at the top of the slope and an unscientific drainage arrangement [5–8]. These unfavorable factors will accelerate the sliding of a landslide [8].

Landslide stability evaluation is an important research topic, and stability coefficients assist local authorities with landslide prevention and treatment measures. There are many methods of studying slope stability, mainly including the traditional limit equilibrium method and numerical simulation method. The traditional evaluation method is based on the concept of limit equilibrium [9–15] and strength reduction methods (SRM) [16–19]. Based on the finite element theory, the numerical simulation method is used to study the slope stability [20,21].

In this paper, we researched a medium-sized landslide affected by rain and excavation on the foot of slopes. We investigated the size, cracks, soil layer distribution, topography, groundwater and other aspects of the landslide. In order to reveal the cause of the landslide, we carried out field geological investigations, drilling and indoor experiments. According to the development of cracks on the landslide body, the landslide was identified as the initial sliding stage. Based on the parameters of saturated direct

shear and saturated residual shear of soil samples in the sliding zone, the factor of safety was calculated by the landslide force transmission method and numerical simulation method.

## 2. Duanjiagou Landslide

### 2.1. The Background of the Duanjiagou Landslide

The Duanjiagou landslide is located in Yingshan County, Nanchong City, Sichuan Province, China, on the left section of the Bazhong to Guangan expressway K134–K135 segment (As shown in Figure 1).

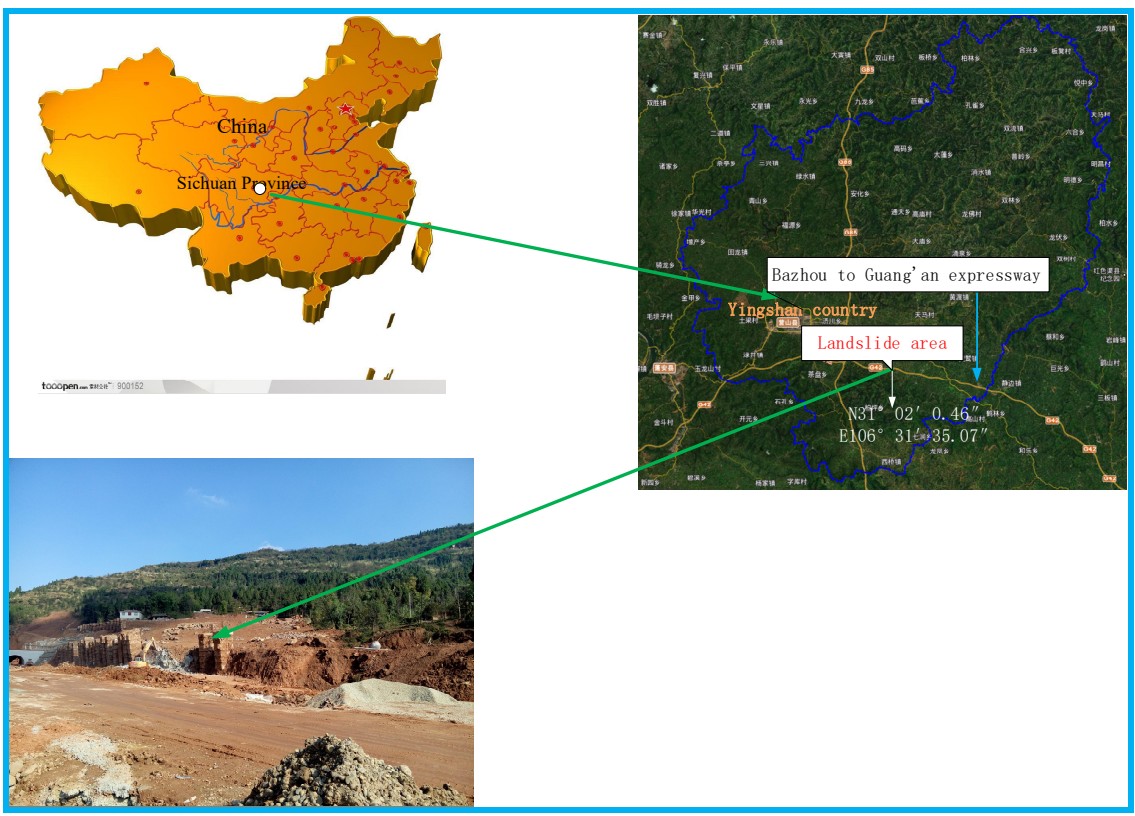

**Figure 1.** Landslide Location. The field is subtropical monsoon climate, and the average annual precipitation is 980–1150 mm.

The average annual temperature is 17.3 °C and the monthly averages range from 5 (January) to 28 °C (July). The highest temperatures occur in July and August, and are characterized by abundant precipitations (Figure 2). Rainfall is not evenly distributed in all seasons, accounting for 45% of the whole year in summer, 25% in spring and autumn, and 5% in winter.

The geomorphology of the landslide site is high in the east and low in the west, and belongs to tectonic denudation mound topography with a slope of 20°–30°. Surface water in the site can be divided into two types. The first type is the stream in the gully, which is smaller during normal times and larger after a rainstorm. The second is the irrigation water in the rice field.

The anti-slide pile is 2 m long and 1 m wide, and the long side direction is parallel to the sliding direction of the landslide, with a depth of 30 m. Due to the excavation on the foot of the slopes, a vertical free surface is formed at the foot of the slope. The slope had a serious sliding deformation on the morning of 4 November 2015. Severe cracking of houses and rural roads in the slope, the No. 40# and 41# anti-slide piles collapsed and broke and the No. 42# and 43# anti-slide piles inclined substantially. The adjacent anti-slide piles against No. 40# and 43# also showed different degrees of deformation (Figure 3). After the landslide, the construction unit backfilled the excavated roadbed to prevent the landslide from further development.

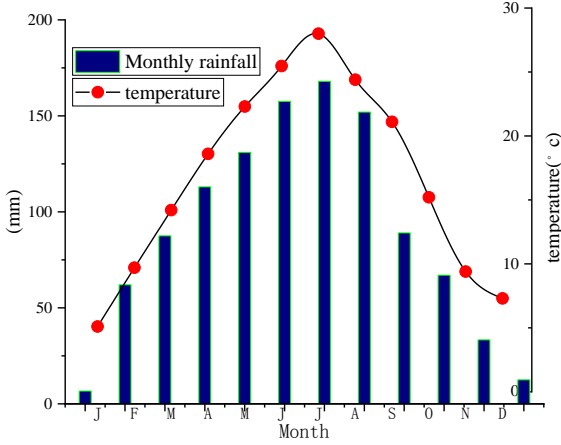

**Figure 2.** Rainfall and temperature distribution in the Yingshan County.

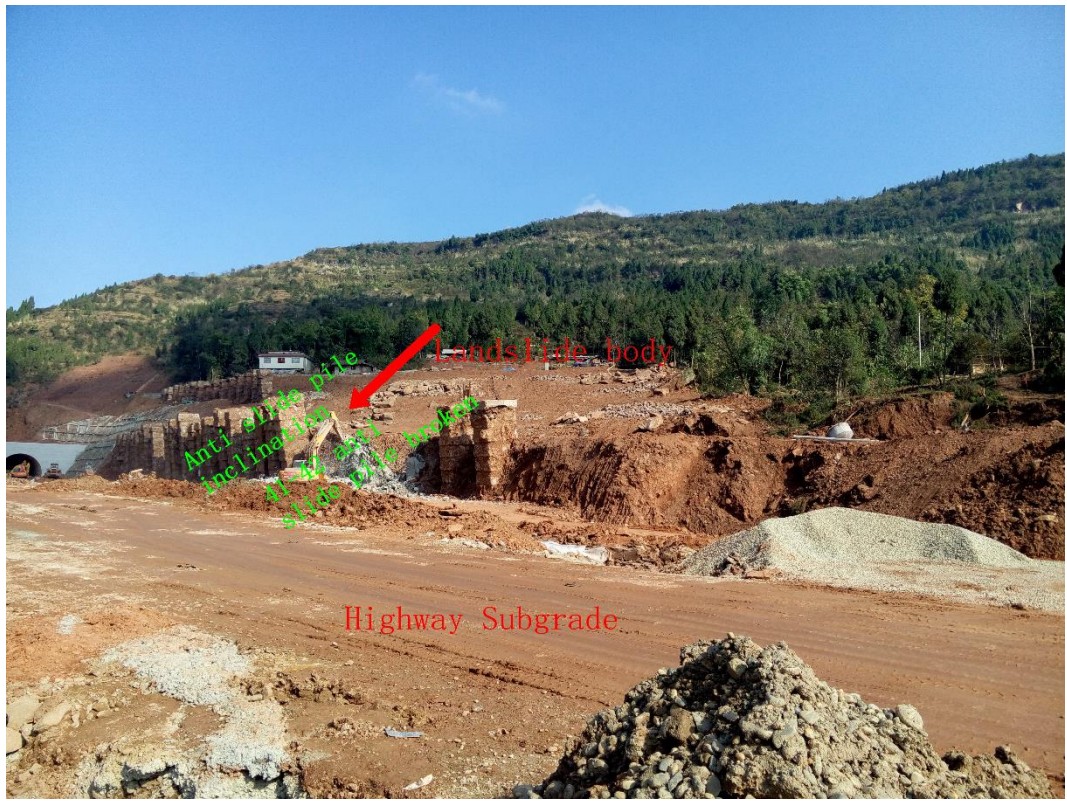

**Figure 3.** Fracture and inclination of the anti-slide pile.

## 2.2. Landslide Features

According to the Classification of Landslide movement, the Landslide is Sliding. The Sliding Surface is Flat and Multi Slide, and the Sliding Soil is Fine-Grained Clay (Cruden and Varnes 1996).

The front elevation of the landslide is 424 m and the back edge elevation is 465 m, forming a relative height difference of 41 m. It is a retrogressive landslide with a sliding direction of 225 degrees. The width of the front edge is 148 m, the length of the main sliding direction is 170 m, the area of the landslide is 16,000 m$^2$, the thickness of the sliding body is 3.1–16.2 m, and the scale of the sliding body is about 192,000 m$^3$. It is a medium-sized landslide.

At the back edge of the landslide, local bedrock is exposed, the dip angle is gentle and the rock mass is stable. The front edge of the landslide is located at the foot of the excavation slope of the

highway roadbed, and an obvious shear outlet can be seen (Figure 4a). The landslide had clear borders and the left side was located in the gully. From the overall shape, the landslide was in the shape of a circle chair.

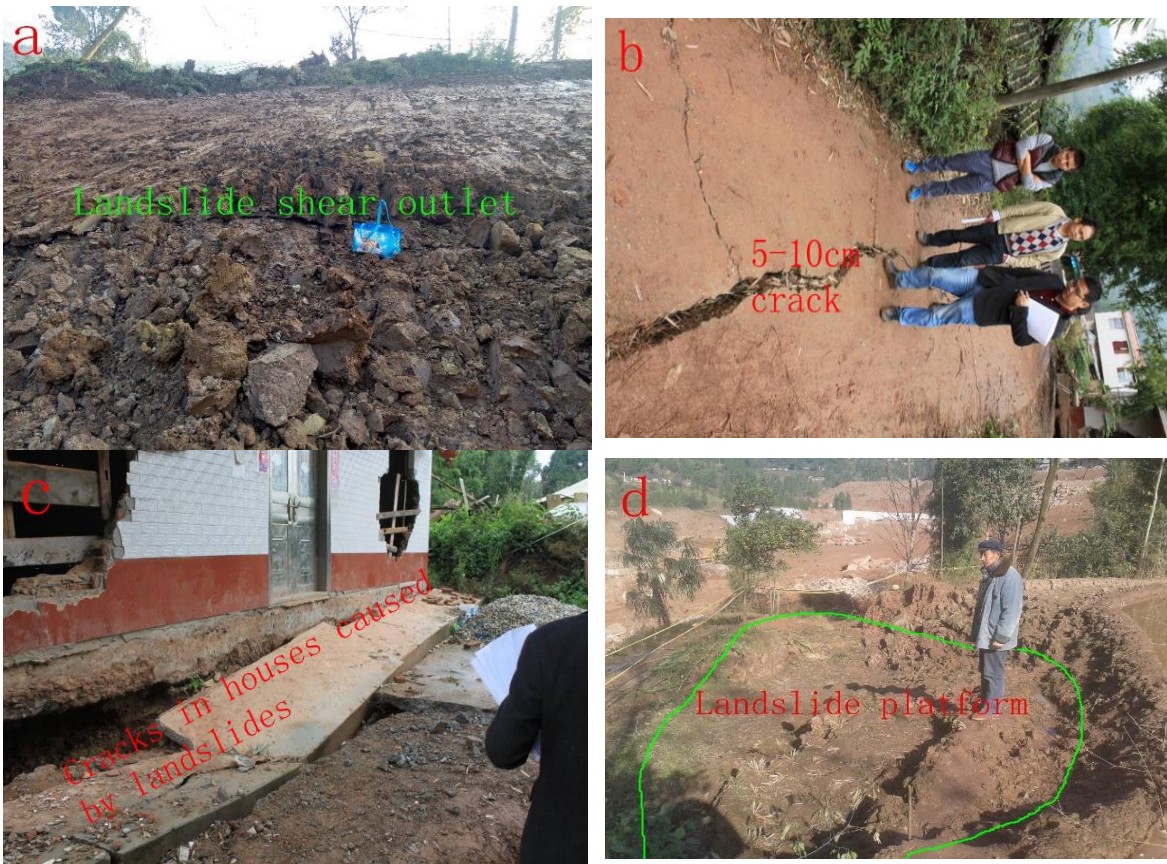

**Figure 4.** (**a**) Shear outlet in front of landslide, (**b**) rural road cracks on the back of the landslide, (**c**) cracks in the building foundation on the boundary of the landslide, (**d**) landslide platform on slide body.

Tensile cracks are developed on the rural road and the upper slope at the back edge of the landslide. Fracture zones are formed with a width of about 1–2 m in local areas, the width of ground cracks varying from 5 to 10 cm (Figure 4b). The house near the boundary was badly cracked (Figure 4c). Due to the deformation, the landslide platform was formed locally with a height of 1 m. (Figure 4d).

## 3. Geotechnical Features of the Landslide

In order to understand the landslide situation better and provide geological data for the slope treatment, we carried out detailed engineering geological investigations. In the landslide survey, three profile lines and nine boreholes were set up, and BZK5, BZK6 boreholes were added to understand the geological conditions in detail. Locations are shown in Figure 5.

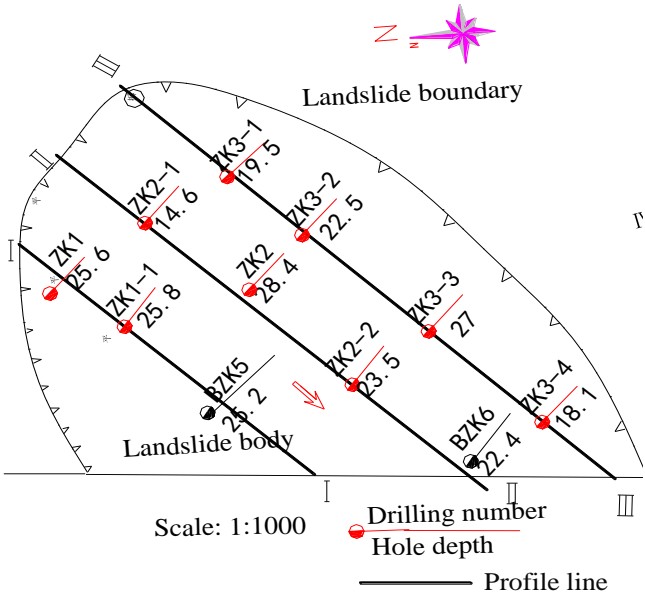

**Figure 5.** Plane distribution of boreholes.

### 3.1. Geotechnical Characteristics of the Soil

The stratum lithology of the landslide body is composed of Quaternary Holocene silty clay, partly mixed with gravel, and the underlying bedrock is $J_{2S}$ mudstone. Silty clay is yellow-brown, widely distributed in sliding bodies, and wet and hard plastic-plastic states. The silty clay in the contact zone with the bedrock surface is relatively soft with large water content, and sandstone fragments can be seen locally.

During the field geological investigations and drilling process, 25 groups of undisturbed soil samples were collected, including 8 groups of slip zone soil samples. Laboratory tests were carried out on soil samples, including density, dry density, specific gravity, void ratio, liquid-plastic limit, water content, saturation and so on. Table 1 lists the properties of the soils.

**Table 1.** Physical properties of landslide soil.

| Natural unit weight | Dry unit weight | Soil particle gravity | Natural porosity ratio | Plastic limit | Liquid limit | Plasticity index |
|---|---|---|---|---|---|---|
| | (kN/m$^3$) | | | (/) | | |
| 18.8–20.1 | 15.4–16.9 | 2.71–2.73 | 0.57–0.726 | 17.8–22.2% | 26.8–36.1 | 10.2–14.6 |
| **Liquid index** | **Saturation** | **Natural moisture content** | **Compression coefficient (0.1–0.2MPa)** | **Compression coefficient (0.2–0.3MPa)** | **Compression modulus (0.1–0.2MPa)** | **Compression modulus (0.2–0.3MPa)** |
| | (%) | | | (MPa$^{-1}$) | | |
| 0.01–0.28 | 0.78–0.95 | 0.137–0.249 | 0.14–0.5 | 0.1–0.38 | 2–10.5 | 2.6–10.3 |

### 3.2. Properties and Strength of Sliding Zone Soil

According to field geological survey data and drilling holes, the slip zone soil was silty clay (Figure 6a), with a thickness of 30–70 cm, mostly yellow-brown, partly reddish-brown and grey-green (Figure 6b), saturation and plastic-soft plastic state. Scratches and folds of the landslide could be seen from the boreholes and shear outlets of the sliding zone soil. (Figure 6c).

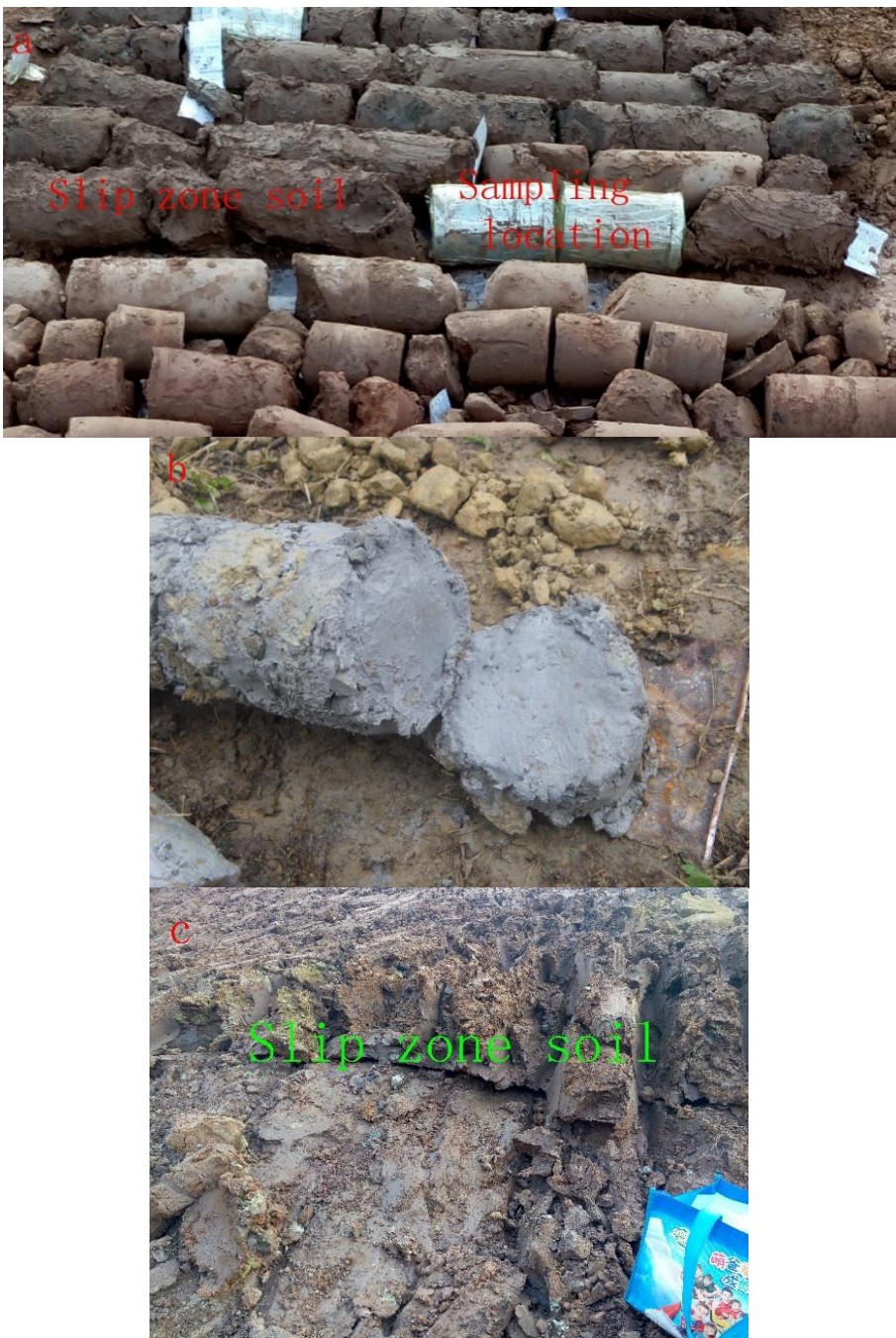

**Figure 6.** (**a**) ZK1-1 borehole sliding zone soil, (**b**) ZK2-2 drilling sliding surface soil, (**c**) sliding zone soil.

From the longitudinal section of landslide sliding, the changing characteristics of the sliding surface can clearly be reflected. The dip angle of the back sliding surface was steeper, and the dip angle of the front sliding surface was smaller. The sliding surface was smooth and generally faced Southwest.

Through borehole sampling, saturated shear tests were carried out on eight undisturbed slip zone soil samples. The mechanical parameters are given in Table 2.

As for the saturated direct shear strength, $c = 15 \sim 51 kPa$ and $\varphi = 3.6 \sim 14.4$; as for the saturated residual shear strength, $c = 14 \sim 47 kPa$ and $\varphi = 3.1 \sim 11.2$. Where the silty clay specimen contained a small amount of gravel, a direct shear test was carried out directly on the specimen.

**Table 2.** Shear strength of slip zone soil.

| Drilling Number | Sampling Depth (m) | Saturated Direct Shear Experiment | | Saturated Residual Shear Experiment | |
|---|---|---|---|---|---|
| | | Cohesion (kPa) | Internal Frictional Angle (°) | Cohesion (kPa) | Internal Frictional Angle (°) |
| ZK1 | 5.7 | 29 | 8 | 15 | 6.5 |
| ZK1-1 | 15.7 | 17 | 12.1 | 14 | 11 |
| ZK2-1 | 2.85 | 34 | 12.5 | 29 | 9.1 |
| ZK2-2 | 9.7 | 17 | 14.4 | 15 | 11.1 |
| ZK3-1 | 6.7 | 51 | 3.6 | 47 | 3.1 |
| ZK3-2 | 9 | 40 | 8.2 | 34 | 6.7 |
| ZK3-3 | 14.6 | 39 | 5 | 29 | 3.3 |
| ZK3-4 | 7.5 | 15 | 12.1 | 14 | 11.2 |

### 3.3. Bedrock

The bedrock of the Duanjiagou landslide was strong weathering and medium weathering mudstone. In this survey, six groups of core samples were taken, where the natural density of the rock was 2.61–2.7 g/cm$^3$, the compressive strength of the natural uniaxial was 11–14.5 MPa, the compressive strength of the saturated single axis was 6.4–8 MPa, and the softening coefficient was 0.33–0.45.

### 3.4. Hydrogeological Conditions of the Landslide

The rainfall in the landslide area was large, and the surface water and groundwater in the slope body was rich. A large number of paddy fields were distributed above the slope, and surface water was heavily infiltrated, resulting in poor engineering performance of the overlying deposits.

Since local clay-mixed gravel strata are porous and loosely structured, surface water is easy to infiltrate. The mudstone layers in the lower part are water-proof layers. Groundwater was mainly stored in silty clay and mixed clay gravel with a water level of 2.5–10.8 m. Due to the seepage of surface water, the soil was saturated and softened at the foot of the slope.

## 4. Landslide Modeling

### 4.1. Standard Value Calculation of Shear Strength

The standard value calculation process of cohesion and internal friction angle is as follows:

$$\text{Average}: \ \varphi_m = \frac{\sum\limits_{i=1}^{n} \varphi_i}{n}. \tag{1}$$

$$\text{Standard deviation}: \ \sigma_f = \sqrt{\frac{\sum\limits_{i=1}^{n} \varphi_i^2 - n * \varphi_m^2}{n-1}}. \tag{2}$$

$$\text{Coefficient}: \ \delta = \frac{\sigma_f}{\varphi_m}. \tag{3}$$

$$\text{Correction factor}: \ \gamma_s = 1 \pm \left( \frac{1.704}{\sqrt{n}} + \frac{4.678}{n^2} \right) \times \delta. \tag{4}$$

$$\text{Standard values}: \ \varphi_k = \gamma_s \times \varphi_m. \tag{5}$$

The standard values of shear strength are obtained by introducing the data in Table 2 into the formulas of mean value, standard deviation, coefficient of variation and correction coefficient. Average value of saturated shear strength is also calculated. The results are shown in Table 3.

**Table 3.** Standard value of saturated direct shear and residual shear.

| Shear Strength Index | Cohesion (kPa) | Internal Frictional Angle (°) |
|---|---|---|
| Saturated direct shear parameters | 19.7 | 6.9 |
| Saturated residual shear parameters | 16.4 | 5.46 |
| Average strength parameters | 18.05 | 6.18 |

### 4.2. Calculation of Safety Factor by Landslide Force Transmission Method

The landslide force transmission method is based on the basic principle of limit equilibrium, and the sliding soil is divided into several soil bars. The force of each block is analyzed, and the glide force is transmitted one by one by using the transfer coefficient.

Basic assumptions: 1. Without considering the extrusion deformation between strips, the landslide slides as a whole and is incompressible; 2. there is no tension crack between the strips and the thrust can only be transmitted; 3. the thrust is expressed as a concentrated force acting at the midpoint of the interface; 4. in the direction of landslide, the soil of unit width is taken as the basic section to calculate, and the frictional force on both sides of the strip is not taken into account. The calculation formula is as follows:

$$F_s = \frac{\sum\limits_{j=1}^{n-1} \left( R_i \prod\limits_{j=1}^{n-1} \psi_j \right) + R_n}{\sum\limits_{j=1}^{n-1} \left( T_i \prod\limits_{j=1}^{n-1} \psi_j \right) + T_n}. \tag{6}$$

$$\psi_j = \cos(\theta_i - \theta_{i+1}) - \sin(\theta_i - \theta_{i+1}) \tan \varphi_{i+1}. \tag{7}$$

$$R_i = N_i \tan \varphi_i + C_i L_i. \tag{8}$$

where $F_s$—Factor of safety

$\theta_i$—The angle between the sliding surface of the $i$ block and the horizontal surface (°)

$R_i$—Sliding force acting on section $i$ (KN/m), $\varphi_i$—Internal friction angle of soil in section $i$ (°)

$C_i$—Cohesion force of soil in section $i$ (Kpa); $L_i$—Slide surface length of section $i$ (m)

$T_i$—Sliding force (KN/m), acting on the sliding surface of the $i$ block, when sliding forces opposite the sliding direction occur, $T_i$ negative values should be taken.

$\psi_j$—Transfer coefficient when the remaining sliding power of the $i$ block is passed to the $i + 1$ block segment, $(j = i)$.

The safety factor of the three profiles were calculated by using the shear strength parameters in Table 3. The calculation process is shown in the Appendix A and the calculation results are shown in Table 4.

**Table 4.** Landslide force transmission method for calculating the safety factor.

| | Profile Lines | Factor of Safety (F$_s$) | | |
|---|---|---|---|---|
| | | I-I | II-II | III-III |
| | Saturated direct shear parameters | 1.05 | 1.13 | 1.14 |
| Landslide force transmission method | Saturated direct residue shear parameters | 0.89 | 0.94 | 0.95 |
| | Average value of shear strength parameters | 0.96 | 1.03 | 1.04 |

According to the deformation characteristics of the landslide body and the calculation results of the safety factor, the evaluation criteria of landslide stability were classified as follows: F$_s$ < 1.0, instability; 1.0 < F$_s$ < 1.05, lack of stability; 1.05 < F$_s$ < 1.15, basic stability; and F$_s$ > 1.15, stability.

The calculation results of the landslide force transmission method showed that the safety factor of profile I-I was the smallest, and the safety factor of III-III profile was the largest.

In the field geological survey, the local sliding body formed a fracture zone with a width of about 1–2 m, which indicated that the landslide lacked stability. The safety factor calculated by the average saturated shear strength was consistent with the field geological survey.

### 4.3. Numerical Simulation of Landslide Stability

Slope stability analysis generally adopts the slice method, including the Sweden slice method, simplified Bishop method, Jane Bulletin method and so on. Due to the complicated calculation process of the slice method and the heavy workload of a manual calculation, commercial software or self-developed calculation programs are usually used in engineering design. GeoStudio is useful for a wide variety of geotechnical problems and was developed by GEO-SLOPE International Ltd. It can be used to simulate both saturated and unsaturated ground-water flows under steady-state as well as transient conditions.

In the calculation of landslide stability, the Sweden slice method or simplified Bishop method was used for the circular slip surface, and Morgenstern—Price method was used for the plane slip surface. In this paper, borehole data showed that the sliding surface was plane, so the Morgenstern—Price method was used to calculate the slope safety factor.

The stability was analyzed by GeoStudio-SlOPE/W, based on the Morgenstern—Price approach of limit equilibrium. According to laboratory test results and parameters of saturated direct shear test, saturated residual shear test and average shear strength, numerical simulation of landslide stability was carried out. The numerical simulation figures are shown in Figure 7.

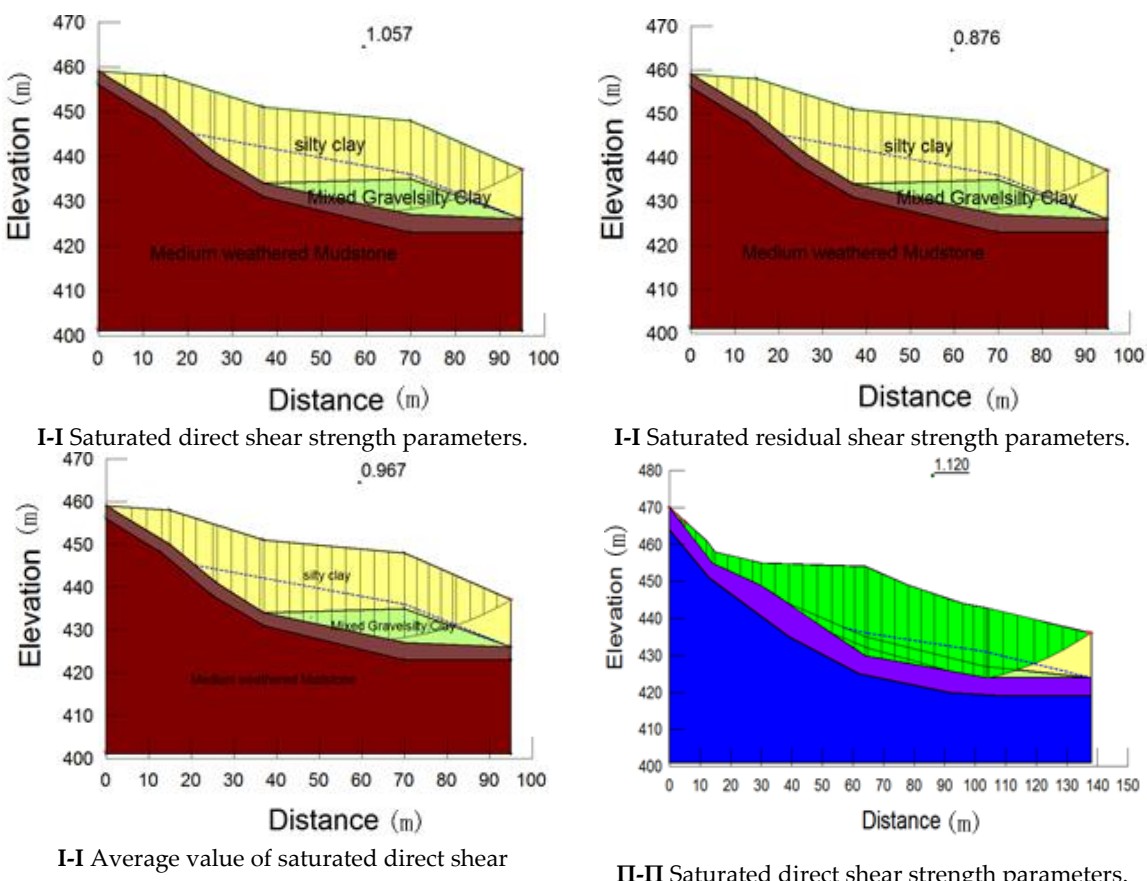

**I-I** Saturated direct shear strength parameters.　　　**I-I** Saturated residual shear strength parameters.

**I-I** Average value of saturated direct shear parameters.　　　**II-II** Saturated direct shear strength parameters.

**Figure 7.** *Cont.*

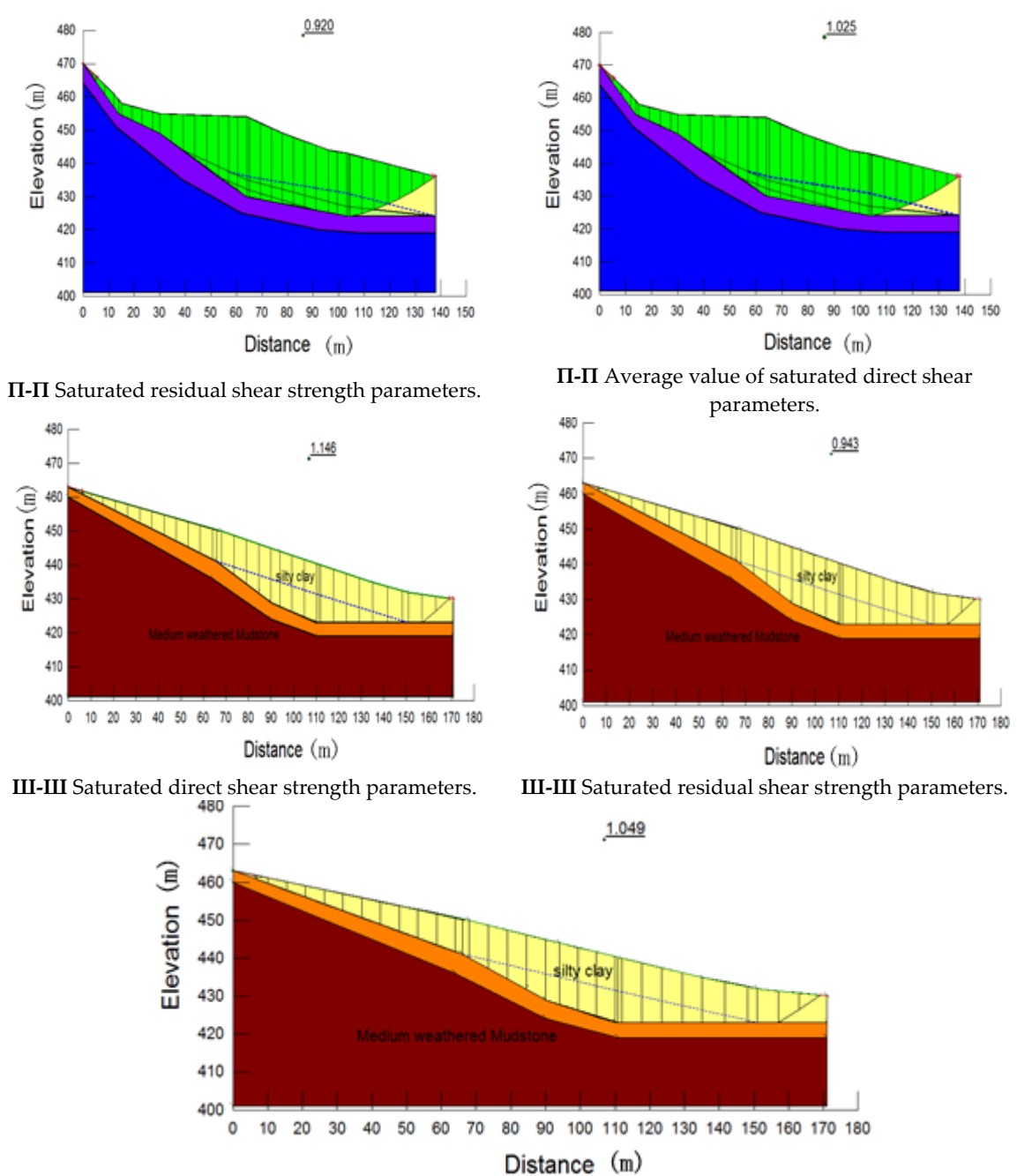

**Π-Π** Saturated residual shear strength parameters.

**Π-Π** Average value of saturated direct shear parameters.

**ΙΙΙ-ΙΙΙ** Saturated direct shear strength parameters.

**ΙΙΙ-ΙΙΙ** Saturated residual shear strength parameters.

**ΙΙΙ-ΙΙΙ** Average value of saturated direct shear parameters.

**Figure 7.** Numerical simulation of landslide stability.

The safety factor ($F_s$) of different profiles were simulated. The numerical simulation results are shown in Table 5.

**Table 5.** Numerical simulation results of the safety coefficient.

| | Profile Lines | Factor of Safety($F_s$) | | |
|---|---|---|---|---|
| | | I-I | Π-Π | ΙΙΙ-ΙΙΙ |
| Numerical simulation of landslide stability | Saturated direct shear parameters | 1.057 | 1.12 | 1.146 |
| | Saturated direct residue shear parameters | 0.876 | 0.92 | 0.943 |
| | Average value of shear strength parameters | 0.967 | 1.025 | 1.049 |

Comparing Table 4 with Table 5, the difference between landslide force transmission method and numerical simulation results was small. The safety factor of the three profiles were different.

When the saturated direct shear parameters were used to calculate the stability, the safety factor was between 1.05 and 1.146, and the landslide was basically stable. When the saturated residual shear parameters were used to calculate the stability, the safety factor was between 0.876 and 0.95, and the landslide was in sliding state. When the average value of saturated shear parameters was used to calculate the stability, the safety factor was between 0.96 and 1.049, and the landslide was in an unstable or lacked a stable state.

For profile I-I of the landslide, the values of the safety factor were quite low, and was the smallest one of the three profiles.

The force transmission method and numerical simulation of the landslide showed that the safety factor calculated by the average saturated shear strength parameters was consistent with the actual situation of the landslide.

## 5. Analysis on the Cause of the Landslide

Before the construction of the anchor frame, the construction company excavated the soil in front of the anti-slide pile, which reduced the anti-sliding force and produced a vertical free surface (Figure 8). This was the main reason for the formation of the landslide.

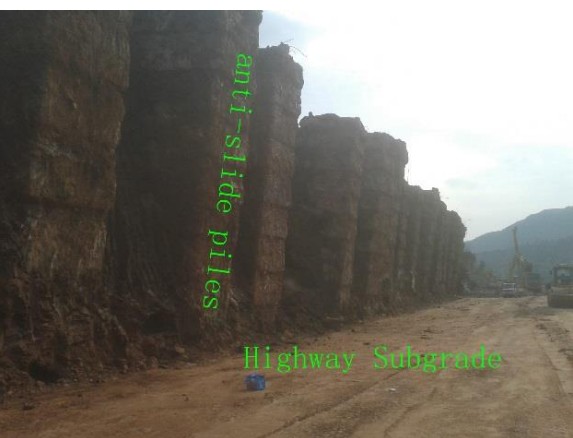

**Figure 8.** Anti-slide pile.

Rainfall is abundant in landslide areas, and there are a large number of paddy fields above the slope. Because a large amount of surface water infiltrates downward, soil properties are poor (Figure 9).

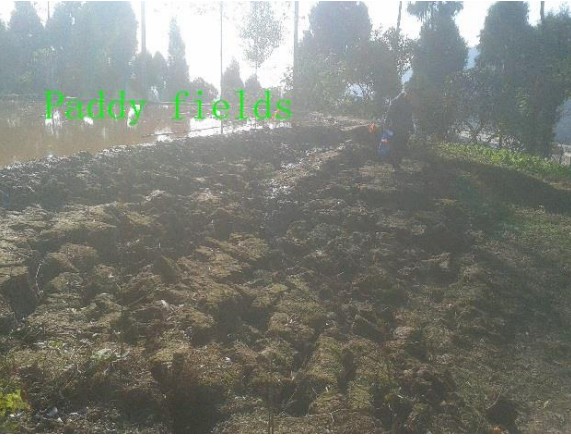

**Figure 9.** Paddy field in upper landslide.

Surface water and groundwater were developed in the slope. The drainage culvert of section K134 + 930 could not drain effectively, which made groundwater gather at the foot of the slope. (Figure 10). The saturation and softening soil gradually reduced the strength of the soil, which caused the failure of the anti-slide piles. The failure of No. 40#–43# anti-slide pile further aggravated the development of the landslide.

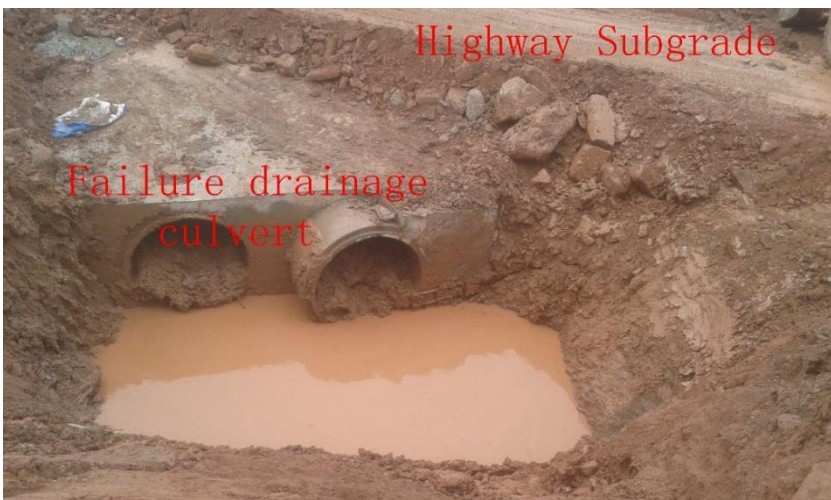

**Figure 10.** Drainage culvert failure at foot of slope.

## 6. Summary and Conclusions

A.W. Skempton (1985) considered that when the clay content in the sliding zone was greater than 30%, the friction factor was controlled by the viscous grain. In this landslide, the slip zone soil was mostly silty clay, partially containing sandstone gravel, and the content was much less than 70%. Low content gravel was scattered in the silty clay, therefore, slope stability was controlled by silty clay strength.

Landslide stability is related to deformation. In the initial sliding stage, the landslide slides at non-uniform speed. Some slip zone soil produces larger displacement, and the strength of soil reaches saturated residual shear strength. The displacement of some slip zone soil is small or not slipping, and the strength of the soil is close to the saturated direct shear strength. Therefore, in the initial sliding stage, the slope stability should be calculated by using the average of the shear strength parameters.

In the field geological survey, the landslide platform was formed with a height of about 0.5–1 m, which indicated that the landslide lacked stability. The calculation of the landslide safety coefficient by average shear strength parameters was in agreement with the field geological survey.

Based on in situ investigations and stability analysis on the Duanjiagou landslide located at the Yingshan County, Nanchong City, Sichuan Province, China, it was possible to formulate some main conclusions as follows:

- Duanjiagou landslide appeared as a sliding deformation on 4 November 2015.The slip zone soil was silty clay, the slip bed was strongly weathered mudstone, and the thickness of the slip body was 3.1–16.2 m, with a wide range of thickness changes. The scale of the sliding body was about 192,000 m$^3$, which is a medium-sized landslide.
- In this paper, according to the parameters of saturated direct shear strength and saturated residual shear strength, the safety factor of different profiles were calculated by using the landslide force transmission method. The results showed that profile I-I was instability, and profiles II-II and III-III were the lack of stability.
- Using Geo-slope software, the numerical simulation of three profiles was carried out respectively, and the numerical simulation results were in agreement with the calculation results of the landslide force transmission method. Due to the non-integral uniform speed of the landslide, in the initial

sliding stage, the landslide safety factor calculated by the average saturated shear strength was more consistent with the field geological survey.

- At present, the landslide is in the unstable or lack of stability state. If the upper surface water of the slope and atmospheric precipitation continue to seep down, and the soil in the sliding zone is saturated and softened, the landslide will accelerate the decline and cause adverse effects.
- The landslide is harmful to the highway under construction and the residents nearby. Considering the importance of slope stability and landslide prevention, we have suggested that drainage combined with anti-slide pile anchor should be taken in time.

**Author Contributions:** Longju Zhang contributed to the data analysis and manuscript writing. Xingang Wang proposed the main structure of this study. Tangdai Xia, Min Gao and Kang Cheng. provided useful advice and revised the manuscript. All of the authors read and approved the final manuscript. The geological field survey was completed from 16 December 2015 to March 2016. The first author was mainly responsible for the geological field investigation, drilling, sampling and the compiling of survey reports. A number of comrades completed field and indoor work. I would like to express my heartfelt thanks to them. All authors have read and agreed to the published version of the manuscript.

**Acknowledgments:** This work was funded by the National Natural Fund, Multi-field coupled salt-frost heaving damage model of lining channel under water and salt recharge conditions in cold and arid regions, (grant number 51641903) and the University of Tarim President Fund, Experimental study on mechanical properties of aeolian sand in Taklimakan, (grant number TDZKQN201705).

**Conflicts of Interest:** The authors declare no conflicts of interest.

## Appendix A

## Calculation of landslide safety factor by saturated residual shear strength parameters

| Split block | Severe KN/m³ | Area m² | Weight (KN/m) | θi° | sinθi | cosθi | φi | tφφi | ci | li | ci li KN/m | △θ° | cos △θ | sin △θ | Transfer coefficient ψj | $\prod_{j=1}^{n-1} \psi_j$ | Sliding force KN/m | Method to split force KN/m | Anti-slip force KN/m | Total anti-skid force KN/m | Total sliding Force KN/m | factor of Safty (Fs) |
|---|---|---|---|---|---|---|---|---|---|---|---|---|---|---|---|---|---|---|---|---|---|---|
| | | | | | | | | | | | | | | | | | | | | | | |
| Ⅰ - Ⅰ 'Longitudinal profile stability calculation table | | | | | | | | | | | | | | | | | | | | | | |
| 1 | 20 | 46.2 | 923 | 25 | 0.429 | 0.903 | 5.46 | 0.096 | 16.4 | 15.6 | 256.496 | -15.0 | 0.966 | -0.259 | 0.991 | 0.866 | 395.9 | 833.8 | 336.2 | 5705.693 | 6435.633 | 0.89 |
| 2 | 20 | 101.2 | 2024.5 | 40 | 0.648 | 0.762 | 5.46 | 0.096 | 16.4 | 13.8 | 226.976 | 8.7 | 0.988 | 0.151 | 0.974 | 0.874 | 1312.1 | 1541.7 | 374.3 | | | |
| 3 | 20 | 188.5 | 3769.75 | 32 | 0.525 | 0.851 | 5.46 | 0.096 | 16.4 | 13.3 | 218.12 | 15.3 | 0.965 | 0.264 | 0.939 | 0.897 | 1980.9 | 3207.3 | 524.7 | | | |
| 4 | 20 | 360.5 | 7209.5 | 16 | 0.282 | 0.959 | 5.46 | 0.096 | 16.4 | 18.0 | 295.2 | 6.8 | 0.993 | 0.118 | 0.982 | 0.955 | 2035.5 | 6916.2 | 956.3 | | | |
| 5 | 20 | 342.9 | 6858.5 | 10 | 0.167 | 0.986 | 5.46 | 0.096 | 16.4 | 16.0 | 261.826 | 8.9 | 0.988 | 0.155 | 0.973 | 0.973 | 1143.8 | 6762.5 | 908.2 | | | |
| 6 | 20 | 453.1 | 9061 | 1 | 0.012 | 1.000 | 5.46 | 0.096 | 16.4 | 27.6 | 453.132 | | | | 1.000 | 1.000 | 110.7 | 9060.3 | 1319.2 | | | |
| Ⅱ - Ⅱ 'Longitudinal profile stability calculation table | | | | | | | | | | | | | | | | | | | | | | |
| 1 | 20 | 55.8 | 1115.8 | 27 | 0.460 | 0.888 | 5.46 | 0.096 | 16.4 | 32.7 | 535.87 | -11.4 | 0.980 | -0.198 | 0.999 | 0.861 | 513.5 | 990.6 | 630.6 | 7234.164 | 7685.021 | 0.94 |
| 2 | 20 | 189.0 | 3780.5 | 39 | 0.627 | 0.779 | 5.46 | 0.096 | 16.4 | 21.7 | 355.142 | 12.4 | 0.977 | 0.215 | 0.956 | 0.861 | 2368.9 | 2946.3 | 636.8 | | | |
| 3 | 20 | 346.9 | 6938 | 26 | 0.445 | 0.896 | 5.46 | 0.096 | 16.4 | 19.4 | 318.57 | 18.1 | 0.951 | 0.311 | 0.921 | 0.901 | 3084.9 | 6214.4 | 912.6 | | | |
| 4 | 20 | 815.4 | 16307.5 | 8 | 0.144 | 0.990 | 5.46 | 0.096 | 16.4 | 38.8 | 636.566 | 7.7 | 0.991 | 0.134 | 0.978 | 0.978 | 2354.1 | 16136.7 | 2179.0 | | | |
| 5 | 20 | 581.0 | 11620.5 | 1 | 0.010 | 1.000 | 5.46 | 0.096 | 16.4 | 35.3 | 579.084 | | | | 1.000 | 1.000 | 121.7 | 11619.9 | 1689.8 | | | |
| Ⅲ-Ⅲ"Longitudinal profile stability calculation table | | | | | | | | | | | | | | | | | | | | | | |
| 1 | 20 | 228.0 | 4559 | 19 | 0.326 | 0.946 | 5.46 | 0.096 | 16.4 | 45.3 | 742.92 | 1.8 | 1.000 | 0.031 | 0.997 | 0.879 | 1484.3 | 4310.6 | 1154.9 | 7968.738 | 8426.396 | 0.95 |
| 2 | 20 | 288.1 | 5762.5 | 17 | 0.296 | 0.955 | 5.46 | 0.096 | 16.4 | 26.8 | 439.602 | -12.8 | 0.975 | -0.222 | 0.996 | 0.882 | 1704.0 | 5504.8 | 965.8 | | | |
| 3 | 20 | 354.8 | 7096.5 | 30 | 0.500 | 0.866 | 5.46 | 0.096 | 16.4 | 24.6 | 403.358 | 13.2 | 0.974 | 0.228 | 0.952 | 0.885 | 3548.3 | 6145.7 | 990.8 | | | |
| 4 | 20 | 461.6 | 9231.5 | 17 | 0.289 | 0.957 | 5.46 | 0.096 | 16.4 | 22.8 | 374.248 | 16.8 | 0.957 | 0.289 | 0.930 | 0.930 | 2668.2 | 8837.5 | 1219.0 | | | |
| 5 | 20 | 752.3 | 15046 | 0 | 0.000 | 1.000 | 5.46 | 0.096 | 16.4 | 40.1 | 657.64 | 0.0 | 1.000 | 0.000 | 1.000 | 1.000 | 0.0 | 15046.0 | 2095.8 | | | |
| 6 | 20 | 226.3 | 4525 | 0 | 0.000 | 1.000 | 5.46 | 0.096 | 16.4 | 22.2 | 364.408 | | | | 1.000 | 1.000 | 0.0 | 4525.0 | 796.9 | | | |

**Figure A1.** *Cont.*

Calculating the safety factor of landslide by average value of shear strength parameters

| Split block | Severe KN/m³ | area m² | weight KN/m | inclination and number of culvert | | | friction resistance | | viscosity resistance | | | Inclination and function | | | transfer coefficient ψⱼ | $\prod_{j=1}^{n-1} \psi_j$ | sliding force KN/m | method to split force KN/m | anti-slip force KN/m | Total anti-skid force KN/m | Total sliding Force KN/m | factor of Safty ( Fs) |
|---|---|---|---|---|---|---|---|---|---|---|---|---|---|---|---|---|---|---|---|---|---|---|
| | | | | $\theta i°$ | $\sin\theta i$ | $\cos\theta i$ | $\varphi i$ | $ty\varphi i$ | $c i$ | $l i$ | $c i\, l i$ KN/m | $\triangle\theta°$ | $\cos \triangle\theta$ | $\sin \triangle\theta$ | | | | | | | | |
| Ⅰ - Ⅰ 'Longitudinal profile stability calculation table | | | | | | | | | | | | | | | | | | | | | | |
| 1 | 20 | 46.2 | 923 | 25 | 0.429 | 0.903 | 6.18 | 0.108 | 18.05 | 15.6 | 282.302 | -18.0 | 0.951 | -0.309 | 0.985 | 0.840 | 395.9 | 833.8 | 372.6 | 6178.103 | 6443.030 | 0.96 |
| 2 | 20 | 101.2 | 2024.5 | 43 | 0.687 | 0.727 | 6.18 | 0.108 | 18.05 | 13.8 | 249.812 | 11.7 | 0.979 | 0.203 | 0.957 | 0.853 | 1391.0 | 1471.0 | 409.1 | | | |
| 3 | 20 | 188.5 | 3769.75 | 32 | 0.525 | 0.851 | 6.18 | 0.108 | 18.05 | 13.3 | 240.065 | 15.3 | 0.965 | 0.264 | 0.936 | 0.891 | 1980.9 | 3207.3 | 587.4 | | | |
| 4 | 20 | 360.5 | 7209.5 | 16 | 0.282 | 0.959 | 6.18 | 0.108 | 18.05 | 18.0 | 324.9 | 6.8 | 0.993 | 0.118 | 0.980 | 0.952 | 2035.5 | 6916.2 | 1073.8 | | | |
| 5 | 20 | 342.9 | 6858.5 | 10 | 0.167 | 0.986 | 6.18 | 0.108 | 18.05 | 16.0 | 288.1683 | 8.9 | 0.988 | 0.155 | 0.971 | 0.971 | 1143.8 | 6762.5 | 1020.4 | | | |
| 6 | 20 | 453.1 | 9061 | 1 | 0.012 | 1.000 | 6.18 | 0.108 | 18.05 | 27.6 | 498.7215 | | | | 1.000 | 1.000 | 110.7 | 9060.3 | 1479.8 | | | |
| Ⅱ - Ⅱ 'Longitudinal profile stability calculation table | | | | | | | | | | | | | | | | | | | | | | |
| 1 | 20 | 55.8 | 1115.8 | 27 | 0.460 | 0.888 | 6.18 | 0.108 | 18.05 | 32.7 | 589.7838 | -11.4 | 0.980 | -0.198 | 1.002 | 0.855 | 513.5 | 990.6 | 697.1 | 7904.240 | 7643.435 | 1.03 |
| 2 | 20 | 189.0 | 3780.5 | 39 | 0.627 | 0.779 | 6.18 | 0.108 | 18.05 | 21.7 | 390.8728 | 12.4 | 0.977 | 0.215 | 0.953 | 0.854 | 2368.9 | 2946.3 | 709.9 | | | |
| 3 | 20 | 346.9 | 6938 | 26 | 0.445 | 0.896 | 6.18 | 0.108 | 18.05 | 19.4 | 350.6213 | 18.1 | 0.951 | 0.311 | 0.917 | 0.895 | 3084.9 | 6214.4 | 1023.5 | | | |
| 4 | 20 | 815.4 | 16307.5 | 8 | 0.144 | 0.990 | 6.18 | 0.108 | 18.05 | 38.8 | 700.6108 | 7.7 | 0.991 | 0.134 | 0.976 | 0.976 | 2354.1 | 16136.7 | 2447.9 | | | |
| 5 | 20 | 581.0 | 11620.5 | 1 | 0.010 | 1.000 | 6.18 | 0.108 | 18.05 | 35.3 | 637.3455 | | | | 1.000 | 1.000 | 121.7 | 11619.9 | 1895.6 | | | |
| Ⅲ-Ⅲ"Longitudinal profile stability calculation table | | | | | | | | | | | | | | | | | | | | | | |
| 1 | 20 | 228.0 | 4559 | 19 | 0.326 | 0.946 | 6.18 | 0.108 | 18.05 | 45.3 | 817.665 | 1.8 | 1.000 | 0.031 | 0.996 | 0.874 | 1484.3 | 4310.6 | 1284.4 | 8757.679 | 8382.440 | 1.04 |
| 2 | 20 | 288.1 | 5762.5 | 17 | 0.296 | 0.955 | 6.18 | 0.108 | 18.05 | 26.8 | 483.8303 | -12.8 | 0.975 | -0.222 | 0.999 | 0.878 | 1704.0 | 5504.8 | 1079.9 | | | |
| 3 | 20 | 354.8 | 7096.5 | 30 | 0.500 | 0.866 | 6.18 | 0.108 | 18.05 | 24.6 | 443.9398 | 13.2 | 0.974 | 0.228 | 0.949 | 0.879 | 3548.3 | 6145.7 | 1109.4 | | | |
| 4 | 20 | 461.6 | 9231.5 | 17 | 0.289 | 0.957 | 6.18 | 0.108 | 18.05 | 22.8 | 411.901 | 16.8 | 0.957 | 0.289 | 0.926 | 0.926 | 2668.2 | 8837.5 | 1368.8 | | | |
| 5 | 20 | 752.3 | 15046 | 0 | 0.000 | 1.000 | 6.18 | 0.108 | 18.05 | 40.1 | 723.805 | 0.0 | 1.000 | 0.000 | 1.000 | 1.000 | 0.0 | 15046.0 | 2353.0 | | | |
| 6 | 20 | 226.3 | 4525 | 0 | 0.000 | 1.000 | 6.18 | 0.108 | 18.05 | 22.2 | 401.071 | | | | 1.000 | 1.000 | 0.0 | 4525.0 | 891.0 | | | |

**Figure A1.** *Cont.*

Calculation of landslide safety factor by saturated direct shear strength parameters

| Split block | Severe KN/m³ | Area m² | Weight KN/m | Inclination and number of culvert | | | Friction resistance | | Viscosity resistance | | | Inclination and function | | | Transfer coefficient ψj | $\prod_{j=1}^{n-1}\psi_j$ | Sliding force KN/m | Method to split force KN/m | Anti-slip force KN/m | Total anti-skid force KN/m | Total sliding Force KN/m | factor of Safty (Fs) |
|---|---|---|---|---|---|---|---|---|---|---|---|---|---|---|---|---|---|---|---|---|---|---|
| | | | | θi° | sinθi | cosθi | φi | tγφi | ci | li | ci li KN/m | △θ° | cos△θ | sin△θ | | | | | | | | |
| I - I 'Longitudinal profile stability calculation table | | | | | | | | | | | | | | | | | | | | | | |
| 1 | 20 | 46.2 | 923 | 25 | 0.429 | 0.903 | 6.9 | 0.121 | 19.7 | 15.6 | 308.108 | -15.0 | 0.966 | -0.259 | 0.997 | 0.856 | 395.9 | 833.8 | 409.0 | 6688.283 | 6367.407 | 1.05 |
| 2 | 20 | 101.2 | 2024.5 | 40 | 0.648 | 0.762 | 6.9 | 0.121 | 19.7 | 13.8 | 272.648 | 8.7 | 0.988 | 0.151 | 0.970 | 0.858 | 1312.1 | 1541.7 | 459.2 | | | |
| 3 | 20 | 188.5 | 3769.75 | 32 | 0.525 | 0.851 | 6.9 | 0.121 | 19.7 | 13.3 | 262.01 | 15.3 | 0.965 | 0.264 | 0.933 | 0.885 | 1980.9 | 3207.3 | 650.1 | | | |
| 4 | 20 | 360.5 | 7209.5 | 16 | 0.282 | 0.959 | 6.9 | 0.121 | 19.7 | 18.0 | 354.6 | 6.8 | 0.993 | 0.118 | 0.979 | 0.949 | 2035.5 | 6916.2 | 1191.5 | | | |
| 5 | 20 | 342.9 | 6858.5 | 10 | 0.167 | 0.986 | 6.9 | 0.121 | 19.7 | 16.0 | 314.5105 | 8.9 | 0.988 | 0.155 | 0.969 | 0.969 | 1143.8 | 6762.5 | 1132.9 | | | |
| 6 | 20 | 453.1 | 9061 | 1 | 0.012 | 1.000 | 6.9 | 0.121 | 19.7 | 27.6 | 544.311 | | | | 1.000 | 1.000 | 110.7 | 9060.3 | 1640.7 | | | |
| II - II 'Longitudinal profile stability calculation table | | | | | | | | | | | | | | | | | | | | | | |
| 1 | 20 | 55.8 | 1115.8 | 27 | 0.460 | 0.888 | 6.9 | 0.121 | 19.7 | 32.7 | 643.6975 | -11.4 | 0.980 | -0.198 | 1.004 | 0.850 | 513.5 | 990.6 | 763.6 | 8571.604 | 7601.880 | 1.13 |
| 2 | 20 | 189.0 | 3780.5 | 39 | 0.627 | 0.779 | 6.9 | 0.121 | 19.7 | 21.7 | 426.6035 | 12.4 | 0.977 | 0.215 | 0.951 | 0.846 | 2368.9 | 2946.3 | 783.1 | | | |
| 3 | 20 | 346.9 | 6938 | 26 | 0.445 | 0.896 | 6.9 | 0.121 | 19.7 | 19.4 | 382.6725 | 18.1 | 0.951 | 0.311 | 0.913 | 0.890 | 3084.9 | 6214.4 | 1134.7 | | | |
| 4 | 20 | 815.4 | 16307.5 | 8 | 0.144 | 0.990 | 6.9 | 0.121 | 19.7 | 38.8 | 764.6555 | 7.7 | 0.991 | 0.134 | 0.975 | 0.975 | 2354.1 | 16136.7 | 2717.4 | | | |
| 5 | 20 | 581.0 | 11620.5 | 1 | 0.010 | 1.000 | 6.9 | 0.121 | 19.7 | 35.3 | 695.607 | | | | 1.000 | 1.000 | 121.7 | 11619.9 | 2101.8 | | | |
| III-III"Longitudinal profile stability calculation table | | | | | | | | | | | | | | | | | | | | | | |
| 1 | 20 | 228.0 | 4559 | 19 | 0.326 | 0.946 | 6.9 | 0.121 | 19.7 | 45.3 | 892.41 | 1.8 | 1.000 | 0.031 | 0.996 | 0.870 | 1484.3 | 4310.6 | 1414.1 | 9543.547 | 8338.401 | 1.14 |
| 2 | 20 | 288.1 | 5762.5 | 17 | 0.296 | 0.955 | 6.9 | 0.121 | 19.7 | 26.8 | 528.0585 | -12.8 | 0.975 | -0.222 | 1.002 | 0.874 | 1704.0 | 5504.8 | 1194.2 | | | |
| 3 | 20 | 354.8 | 7096.5 | 30 | 0.500 | 0.866 | 6.9 | 0.121 | 19.7 | 24.6 | 484.5215 | 13.2 | 0.974 | 0.228 | 0.946 | 0.872 | 3548.3 | 6145.7 | 1228.2 | | | |
| 4 | 20 | 461.6 | 9231.5 | 17 | 0.289 | 0.957 | 6.9 | 0.121 | 19.7 | 22.8 | 449.554 | 16.8 | 0.957 | 0.289 | 0.922 | 0.922 | 2668.2 | 8837.5 | 1519.0 | | | |
| 5 | 20 | 752.3 | 15046 | 0 | 0.000 | 1.000 | 6.9 | 0.121 | 19.7 | 40.1 | 789.97 | 0.0 | 1.000 | 0.000 | 1.000 | 1.000 | 0.0 | 15046.0 | 2610.7 | | | |
| 6 | 20 | 226.3 | 4525 | 0 | 0.000 | 1.000 | 6.9 | 0.121 | 19.7 | 22.2 | 437.734 | | | | 1.000 | 1.000 | 0.0 | 4525.0 | 985.3 | | | |

**Figure A1.** The calculation process of safety factors.

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
