# Peer review of "Field Geological Investigations and Stability Analysis of Duanjiagou Landslide"

_ijgi, doi:10.3390/ijgi9010023_

Round 1

Reviewer 1 Report

In this manuscript, the shear strength of the slip surface was obtained by back analysis based on the field investigation and soil element tests such as direct shear test and residual shear test. There are some scientific concern about it.

1) There are no describe on the detail of the shear tests. How was the saturation degress of the specimen confirmed?

2) There are no figures on the results of soil element tests.

3) The difference between the peak strength by the direct shear test and the residual strength by residual shear test is very small. 

4) The difference may be due to test variation. 

5) How the transfer coefficient is determined?

6) There are no Appendix 1 in the manuscript.

7) The slip surface obtained by the numerical simulation is coincident the field investigation result?

8) The sentence of "When the anchor frame is not implemented, the construction unit excavated the soil in front of the pile." on page 11 is difficult to catch the meaning.

9) The drainage combined with anti-slide pile anchor is suggested in the conclusions, but the authority is not shown in the manuscript.

10) The figures shown in Fig.6 must be much clearer.

11) The detail of the anti-slide pile such as the shape, length, diameter, embedment length and material etc. must be shown based on the field investigation.

Reviewer 2 Report

The paper " Field geological investigations and stability analysis of Duanjiagou landslide, Yingshan County, Nanchong City, Sichuan Province, China” is an interesting study dealing with an extensive landslide. The work is a traditional landslide study.

Globally, the paper is quite clear and well structured but it needs some improvements.

Several figures of the article are not well drawn and have to be changed

If it is possible, I suggest to add a short description of weather and a new figure.

Reference list is complete, although most of references are local.

Comments are listed below.

TITLE

CHANGE THE FINAL PART OF THE TITLE. The final part of the title is too long. Too many locations that are not important for an international audience. You can list them inside the article.

ABSTRACT

General comment

Abstract is well written and clear.

Other comments

Landsliding instead of landslide instability INTRODUCTION

General comment

Introduction has to be change in the final part. I think Authors have to introduce the investigated landslide using a traditional classification such as Cruden and Varnes. Please insert this and a general description of the size of its. This part can be inserted after “ In this paper” and write like “In this paper, we investigate a large landslide affecting…..”

Minor comments

Slope stability evaluation is not a problem!! Change…. Stability coefficients (use plural) assist local authorites ….. instead of an important influence THE INTRODUCTION OF DUANJIAGOU LANDSLIDE

DELETE THE INTRODUCTION OF

General comments on 2.1

This part has to be better assisted by figures. The description of climate is too poor.

Figure 1 Delete the globe, China is ok, Yinshan map is too small, the photo is ok.

Insert a Figure 2 like you can find in Mantovani, Devoto et al. (2013) in Landslides where there a figure of monthly rainfalls and temperatures. More over add two sentences of rainfalls trends.

General comments on 2.2

The title has to be changed. I suggest “Landslide geomorphology or landslide features”. Moreover, the authors have to start classifying the landslide and then they can classify the geomorphological features of the landslide. It can be quite the same of the introduction.

Add a map with the main features described in the text. A simple geomorphological map it will be great. Rotate Part b of Figure 3

Other comments

Apertures instead of width of Varying from 5 to 10 cm instead of 5.0-10.0 cm GEOTECHNICAL FEATURES OF THE LANDSLIDE

Figure 4 has to be improved. Change the caption.

This part is well written and well structured

Other comments

“Table 1 lists the properties of the soils” instead of “The experimental …..” MPa instead of Mpa

LANDSLIDE MODELING maybe it is better

This chapter summarizes very well the results of this paper and includes very interesting tables.

Other comments

I think it is better to use Factor Of Safety (FOS) which is widely used in scientific journals.

5 ANALYSES ON THE CAUSE OF LANDSLIDE

This part is well written.

Other comments

Maybe it is better to write : Soil properties are poor (Figure 8)

SUMMARY AND CONCLUSION

No suggestions. This part is well structured.

REFERENCES

Add Cruden and Varnes reference

Round 2

Reviewer 2 Report

Please modify the following things

Graphs of figure 6 miss (m) along x and y axis. ADD please; 2.2. Landslide features instead of landslid feature; There are stiill Mpa instead of MPa. Please verify.

Best wishes
